# PICU Survivorship: Factors Affecting Feasibility and Cohort Retention in a Long-Term Outcomes Study

**DOI:** 10.3390/children9071041

**Published:** 2022-07-13

**Authors:** Sarah A. Sobotka, Emma J. Lynch, Ayesha V. Dholakia, Anoop Mayampurath, Neethi P. Pinto

**Affiliations:** 1Section of Developmental and Behavioral Pediatrics, Department of Pediatrics, The University of Chicago, 950 East 61st Street, Suite 207, Chicago, IL 60637, USA; elynch1@peds.bsd.uchicago.edu; 2Department of Pediatrics, Boston Children’s Hospital, Boston, MA 02115, USA; ayesha.dholakia@childrens.harvard.edu; 3Department of Biostatistics & Medical Informatics, The University of Wisconsin-Madison, Madison, WI 53705, USA; mayampurath@wisc.edu; 4Division of Critical Care Medicine, Department of Anesthesiology and Critical Medicine, Children’s Hospital of Philadelphia, Philadelphia, PA 19104, USA; pintonp@chop.edu

**Keywords:** long-term outcomes, PICU follow-up, neurodevelopment, feasibility, PICU survivors

## Abstract

Our understanding of longitudinal outcomes of Pediatric Intensive Care Unit (PICU) survivors is limited by the heterogeneity of follow-up intervals, populations, and outcomes assessed. We sought to demonstrate (1) the feasibility of longitudinal multidimensional outcome assessment and (2) methods to promote cohort retention. The objective of this presented study was to provide details of follow-up methodology in a PICU survivor cohort and not to present the outcomes at long-term follow-up for this cohort. We enrolled 152 children aged 0 to 17 years admitted to the PICU in a prospective longitudinal cohort study. We examined resource utilization, family impact of critical illness, and neurodevelopment using the PICU Outcomes Portfolio (POP) Survey which included a study-specific survey and validated tools: 1. Functional Status Scale, 2. Pediatric Evaluation of Disability Inventory Computer Adaptive Test, 3. Pediatric Quality of Life Inventory, 4. Strengths and Difficulties Questionnaire, and 5. Vanderbilt Assessment Scales for Attention Deficit-Hyperactivity Disorder. POP Survey completion rates were 89%, 78%, and 84% at 1, 3, and 6 months. Follow-up rates at 1, 2, and 3 years were 80%, 55%, and 43%. Implementing a longitudinal multidimensional outcome portfolio for PICU survivors is feasible within an urban, tertiary-care, academic hospital. Our attrition after one year demonstrates the long-term follow-up challenges in this population. Our findings inform ongoing efforts to implement core outcome sets after pediatric critical illness.

## 1. Introduction

Mortality in the pediatric intensive care unit (PICU) has decreased dramatically in recent decades, with a concomitant increase in morbidity, suggesting that mortality alone is no longer a sufficient metric of the long-term impact of critical illness [1,2,3,4,5,6]. The sequelae of critical illness are diverse; some children exhibit psychological difficulties as a result of traumatic experiences [7,8,9], while others experience cognitive [10,11,12] or physical [13] deficits. Post-intensive care syndrome in pediatrics (PICS-p) refers to this constellation of physical, cognitive, emotional, and social health problems facing PICU survivors and their families [14,15,16]. PICS-p also acknowledges that a child’s pre-illness baseline, including pre-existing neurodevelopmental disabilities [17] and other premorbid conditions, may also influence a child’s susceptibility to the impact of critical illness [18]. 

Our understanding of the extent of PICS-p is obscured by the variability in duration of follow-up, patient populations studied, and outcome measures. Importantly, PICU survivors do not have a medical home, distinguishing them from preterm infants or children with congenital heart disease who have dedicated follow-up clinics, potentially placing them at increased risk for loss to follow-up. Studies have assessed outcomes at discrete follow-up intervals [13,19,20,21,22,23], but few have followed children longitudinally. Other studies have been limited to follow-up of survivors with a specific subset of pathophysiology [24,25,26,27]. In a scoping review of pediatric critical care medicine literature from 1970–2017, over 360 unique instruments were used to evaluate long-term outcomes [28]. The heterogeneity of these findings limits our understanding of overall PICU survivorship, and in response, there has been increasing attention to the need to develop and implement core outcome sets [29,30,31] (i.e., standardized sets of outcomes to be measured and reported as a minimum in all effectiveness trials in a specific health area [32,33,34]). Ultimately, the widespread use of core outcome sets with standardized outcome measures will improve our understanding of PICS-p, allowing for greater comparability across studies. 

However, prior to adaption, the factors governing feasibility of implementing core outcome sets must be considered. Among the 407 studies included in a recent scoping review of PICU long-term outcomes studies, enrollment in observational PICU survivorship studies varied from 61–100%, with even lower enrollment in qualitative studies ranging from 20–90% [28]. Retention at long-term follow-up was also highly variable, ranging from 68–100% [28]. In this prospective, longitudinal study piloting the use of multidimensional outcome measures to understand the frequency and natural history of PICS-p and to underscore the challenges associated with longitudinal follow-up among a heterogeneous group of children who experience critical illness, we also sought to measure changes in healthcare utilization and the familial and socioeconomic impact of critical illness and PICU survivorship. Here, we report our experience with regard to feasibility and cohort retention using a multidimensional outcome portfolio among PICU survivors through 3 years after hospital discharge. 

## 2. Materials and Methods

Patients. Patients were enrolled specifically for this study from the PICU at Comer Children’s Hospital at the University of Chicago over 15 months (June 2017–August 2018) to account for the seasonal variation inherent in pediatric critical illness and injury. All children aged 0 to 17 years admitted to the PICU were eligible; parental presence at the bedside was necessary for consent and enrollment. No competing clinical or research follow-up programs were in existence at the time of study enrollment, and this cohort does not overlap with previously published cohorts of the authorship team [4,17,35,36]. Children were excluded if they were under state custody due to the logistical challenges of obtaining consent from the state for participation in a research study as well as potential flux in custodial situations at follow-up intervals. Non-English speakers were excluded because some instruments were only available in English with limited ability for translation during remote (telephone or e-mail) follow-up. Informed consent was obtained in person from parent(s)/guardians (hereafter, referred to as “parent(s)”), and assent was obtained from children aged 12 years or older. The protocol was approved by the Institutional Review Board at the University of Chicago.

Screening Process. The study team screened the PICU census for eligible patients and confirmed with the clinical team whether the parent could be approached for consent to participate (potentially denied or deferred due to patient acuity). The study team attempted to recruit several times throughout the hospital stay to capture parents. Bedside nurses facilitated opportunities for consent by notifying the study team when parents became available. 

Measures—Enrollment. Measures were selected from existing validated surveys to capture a broad array of functional and neurodevelopmental outcomes. The study team assessed the Functional Status Scale (FSS) score, a validated measure using a five-point scale in each of six domains (mental, sensory, communication, motor, feeding, and respiratory status) from review of the electronic health record (EHR) or parent interview [37]. Parents completed the other survey instruments intended for caregiver informant report directly: 1. the Healthcare and Neurodevelopmental Profile and Family Impact Survey (Impact Survey), 2. The Pediatric Evaluation of Disability Inventory Computer Adaptive Test (PEDI-CAT), 3. the Pediatric Quality of Life Inventory (PedsQL), 4. the Strengths and Difficulties Questionnaire (SDQ), and 5. the National Institute for Children’s Health Quality Vanderbilt Assessment Scales (Vanderbilt). All survey instruments were administered sequentially in a single survey which comprised our PICU Outcomes Portfolio (POP) survey, except for the PEDI-CAT which was administered on an iPad or over the phone. Full details regarding the scoring of the validated instruments are available elsewhere [38,39,40,41].

The Impact Survey, designed by the study investigators, captures patients’ baseline health utilization and the family impact of the child’s illness. Items were cognitively tested with developmental and behavioral pediatric clinic patients prior to study initiation. The final survey length depended on an individual’s responses, but included about 30 questions. The Impact Survey includes modified questions from the National Survey of Children with Special Health Care Needs [42] regarding the child’s health (e.g., “What kind of place does your child go to when he/she is sick or you need advice about his/her health?”). In addition, questions capture the child’s access to and utilization of neurodevelopmental support services (e.g., “Does your child have an IEP (Individualized Educational Program)?”). Family impact is also assessed (e.g., “Has your child’s health conditions caused financial problems for your family?”). Appendix A.

The PEDI-CAT includes 15 items in each of four domains (daily activities, mobility, social/cognitive, and responsibility) to determine the degree of functioning in each category compared to same-aged peers. Adaptive testing maximizes information gathering while minimizing response burden [38].

The PedsQL is a multidimensional tool intended to quantify health-related quality of life in healthy and ill children up to 18 years of age. The PedsQL consists of 23 items in five domains (physical, psychosocial, emotional, social, and school) [39]. This measure was administered to all parents with versions tailored to the age range of the child: 1–12 months, 13–24 months, 2–4 years, 5–7 years, 8–12 years, or 13–18 years. The population mean for parent-proxy reported 81.3 and standard deviation 15.9 for healthy children [43]. 

The SDQ is a brief behavioral screening survey to assess pro-social behavior and psychopathology of children and adolescents aged 4–17 years [40]. Normative parent SDQ scores for U.S. children are 0–11 (for low difficulties, 12–15 for medium difficulties, and 16–40 for high difficulties [44]. 

The Vanderbilt is a tool that parallels signs and symptoms from DSM-IV to help healthcare professionals diagnose ADHD in children [41]. The Vanderbilt was administered to parents of children at least 5 years of age. Normative scores data for total Attention Deficit Hyperactivity Disorder (ADHD) score, ADHD inattentive, ADHD hyperactive, Oppositional Defiant Disorder (ODD), Conduct Disorder, and Anxiety/Depression are 3.4, 1.6, 1.8, 1.1, 0.5, and 0.4, respectively [45].

Data were collected and managed using the Research Electronic Data Capture (REDCap) tool [46]. Study staff administered all surveys and collected demographic information (name, age, and medical record number) on an iPad Pro (2017). 

The EHR was reviewed for demographic characteristics, admission diagnosis, PICU length of stay, hospital length of stay, and vital signs. Clinical data elements were extracted from the Clinical Research Data Warehouse maintained by the Center for Research Informatics at the University of Chicago in order to calculate the severity of illness indices: Pediatric Risk of Mortality (PRISM) [47] and the Pediatric Sequential Organ Failure Assessment Score (pSOFA) [36] scores.

Measures—Discharge. FSS at hospital discharge was again determined from review of the EHR and/or direct interview with the patient’s PICU physicians or nurses. 

Measures—Follow-up. Parents were contacted either via telephone or e-mail, per stated preference and contact information (1–2 telephone numbers [home and mobile] and an e-mail address) provided at the time of enrollment. No public or private data searches were conducted to obtain contact information. The child’s EHR was queried prior to contact to verify that the parent’s telephone number or e-mail had not changed and to ensure that the child had not died in the interval since last contact. 

During any specific data-collection period, the study team determined, a priori, that 5 telephone calls within 1 month were reasonable and not intrusive. Parents were contacted at subsequent follow-up intervals even if they had not completed follow-up from the preceding interval. Parents received a 20 USD gift card for each completed follow-up, with a total potential incentive of 80 USD throughout the first year of follow-up and an additional 20 USD annually through to the 3-year follow-up.

A subset of the POP survey was completed at each follow-up interval to decrease overall response burden: 1 month (Impact Survey, SDQ), 3 months (Vanderbilt and PEDI-CAT), and 6 months (PedsQL and FSS). All components of the POP survey except for the PEDI-CAT were completed at 1 year after hospital discharge and annually through 3 years after discharge. Investigators anticipated that the time burden associated with reading the PEDI-CAT over the phone (typically self-administered by the participant using an iPad application) would deter participation, and thus re-administration of the PEDI-CAT was deferred until the 3-year follow-up. 

Statistical Analyses. The objective of this presented study was to provide details of data acquisition and follow-up in a PICU survivor cohort and not to present the outcomes at long-term follow-up for this cohort. Descriptive statistics were used to characterize the demographic and clinical variables at each follow-up interval. Participant demographic characteristics were compared to the University of Chicago PICU population during the study period. Follow-up and feasibility data were collected and summarized using descriptive statistics. Analyses were performed using R, version 3.6 (R Project for Statistical Computing, Vienna, Austria), with two-sided *p* < 0.05 denoting statistical significance.

## 3. Results

### 3.1. Enrollment

A total of 832 hospitalized patients were screened for enrollment (Figure 1). There were 119 (14.3%) patients who did not meet the eligibility criteria. Of the 713 eligible children, a total of 496 patients were not approached: 228 (32%) due to lack of availability of parents at the bedside, 50 (7%) because the clinical team denied the study team the opportunity to approach parents for consent, and 218 (30.6%) were discharged prior to enrollment opportunity. Thus, of the 217 available parents, 70% (152 parents) provided consent to participate and were enrolled. 

### 3.2. Cohort Characteristics

The University of Chicago Comer Children’s Hospital is an urban, academic, tertiary-care center that admits medical and surgical pediatric patients. Demographic and clinical characteristics of the study cohort are presented in Table 1. The median (range) age of the children was 5.2 (0–17) years, and 58.6% were male. Participants were 57.2% African American (n = 87), 33.6% Caucasian (n = 51), 9.2% “Other” (n = 14), and 9.2% Hispanic (n = 14). At the time of study enrollment, our typical patient population was 54.2% male, 59.8% African American, 26.7% White, and 24% surgical patients. Our overall PICU median length of stay was 2.9 days. Our study patients were similar to all patients admitted to the PICU during the study enrollment period with regard to sex, race, ethnicity, severity of illness, and length of stay. Study participants were older than the overall PICU population. (Appendix A).

Of the enrolled children, the three most common known admission diagnostic categories were pulmonary, surgical, and neurologic (39.5%, 21.7%, and 8.6%, respectively). The median (IQR) PICU length of stay was 3.0 (1.7–8.1) days, and the overall hospital length of stay was 4 (3–9) days. The mean admission pSOFA and PRISM scores were 4.6 (SD 2.4) and 9.2 (SD 4.5). Demographic and clinical characteristics did not vary between patients who did and did not complete follow-up at each time point.

### 3.3. Cohort Retention and Follow-Up

Of the total 152 patients enrolled, 132 parents (88.6%) completed the 1-month follow-up, 115 (77.7%) completed the 3-month follow-up, 123 (83.1%) completed the 6-month follow-up, and 117 (80.1%) completed the 1-year follow-up. (Figure 2) Of the 152 participants, 122 (80.3%) completed a follow-up for at least 3 of the 4 intervals during the first year of follow-up. At 2 years, 79 (54.5%) children completed the follow-up, and 56 (43.4%) completed the 3-year follow-up. Three patients died during the study follow-up, and four patients withdrew consent. Sixteen children were excluded from 3-year follow-up because they were ≥18 years of age and parents were no longer eligible to serve as informants.

### 3.4. Feasibility Factors

Parents provided their preferred contact method, most often the telephone. E-mail reminders were sent to parents who provided an e-mail address as an alternative contact method. While most parents did not complete their follow-up assessments via e-mail, e-mail correspondence enabled scheduling of telephone follow-ups at a convenient time for parents. 

Among the parents who did not complete the follow-up, the median number of follow-up attempts were 6.5 (IQR 4–8) at 1 month, 5 (IQR 4–6) at 3 months, 6 (IQR 2–12) at 6 months, 11 (IQR 6–17.8) at 1 year, 5 (IQR 5–7) at 2 years, and 5 (IQR 5–6) at 3 years. Among the parents who completed the follow-up, the median number of follow-up attempts were 2 (IQR 1–4) at 1 month, 2 (IQR 1–3) at 3 months, 2 (IQR 1–3) at 6 months, 3 (IQR 2–6) at 1 year, 2 (IQR 1–4) at 2 years, and 4 (IQR 3–5) at 3 years. (Table 2). The majority of parents (89–97%) completed the follow-up during weekdays throughout the year. However, study staff attempted follow-ups on weekends with an additional 3–11% of parents. Parents were most often available during daytime hours (7:00 a.m.–4:59 p.m.) with 77–92% of parents completing follow-ups during these hours. Study staff successfully completed follow-ups in the evenings (5:00 p.m.–6:59 a.m.) for 8–24% of parents. The median number of days to completion of a follow-up survey ranged from 9–28 days. However, as many as 19.3% of parents completed their follow-up questions at a subsequent follow-up touchpoint (e.g., completed the 1-month survey at 3 months).

At least five follow-up attempts were made for most children at each interval; however, 13 children had fewer than three attempts documented at one of their follow-up intervals, which may be due to missed opportunities by our study team. With the exception of one child, all were recaptured at later time intervals. We also note that one child with an unusually prolonged hospitalization (>1 year) was inadvertently lost to follow-up because the child’s follow-up times did not coincide with the rest of the cohort. 

## 4. Discussion

A longitudinal follow-up using a multidimensional PICU core-outcome portfolio is feasible for children and palatable to families discharged from the PICU. Our study has significant findings. Enrollment is the first and greatest hurdle. Consent rates among *available* families were favorable. Retention of this cohort of families for one year was high but there was notable attrition at 2 and 3 years. Our study demonstrates feasibility of implementing a core set of multidimensional outcomes that longitudinally assess the physical, cognitive, educational, social, emotional, behavioral, health-related quality of life, and family impact of pediatric critical illness in a heterogeneous patient population. 

Enrollment of parents of PICU patients is challenging; most parents in our urban population were not available for consent and participation. Nearly one-third of eligible parents were not present throughout their child’s PICU hospitalization. The lack of parental presence is likely multifactorial due to lack of flexible employment schedules, caregiving responsibilities for other children, transportation barriers, and financial barriers (for food, parking, etc.). Another 30% of children were discharged prior to an approach by the study team, likely due to short stays or discharge at night or on weekends when the study team was not always available (reflecting the aforementioned logistical and socioeconomic challenges). A small percentage of families—7%—were not approached by the study team because of the care team’s concerns about suitability for a study focused on long-term outcomes due to the patient’s acuity and uncertain prognosis for survival. 

While only 21% of all *eligible* families enrolled, the majority of families that were available to the study team agreed to participate. This high consent rate of 70% indicates a subgroup of patients that is willing to participate in studies of long-term follow-up. Our ability to enroll a smaller number of families compared to those who consented, reflecting loss to follow-up, may be explained in part by the demographic and social characteristics associated with risk of poor outcomes. For example, young caregiver age, caregiver language barriers, presence of social supports, transportation challenges, or caregiver intellectual disability are risk factors for adverse PICS-p outcomes [48]. These same factors may inhibit access to care, precluding participation in routine clinical or research-related follow-up. 

Our data further reveal that maintenance of a cohesive cohort of 78–89% of these families of PICU survivors during the first year after hospital discharge was possible but necessitates strategic cohort-retention techniques with persistent contact attempts, incentives, flexible modalities, and attention to response burden. Our response rates were comparable to established rates of PICU follow-up at the 3-month follow-up interval. Notably, our response rates at the 1-month, 6-month, and 1-year intervals were more favorable than typical follow-up rates despite the lengthier response burden of the enrollment and 3-month instruments [4,23]. The higher response rates at these three intervals suggest that families were not deterred by the lengthier multidimensional surveys at the preceding intervals. Additionally, parents may have been most willing to participate 1 month after hospital discharge due to the proximity of their PICU experience as compared to subsequent follow-up intervals. Although parents remained highly responsive throughout the 1-year follow-up period, our response rates decreased at 2 and 3 years. 

The reasons for our robust follow-up rates and ability to retain the study cohort through 1 year are likely multifactorial. Parents who completed follow-up typically required 2–3 contact attempts, with weekday and daytime hours yielding the highest follow-up rates. Having study staff also available intermittently on weekends and evenings did allow for greater overall cohort retention [49]; notably, the touchpoints during these times were generally brief and not overly burdensome for team members, but occasionally looped in otherwise difficult-to-reach parents. Additionally, our design included multiple attempts to contact parents based on their expressed preferred method of contact, allowed for flexibility regarding scheduling follow-up via telephone or e-mail, and attempted to recapture parents who were “lost to follow-up” at a previous interval. Gift cards for each participation interval may have been an important financial incentive for participation. The majority of parents completed three of the four follow-up assessments over the first year after discharge suggesting that our approach was effective. 

Our attention to response burden may have also had a positive impact on our response rates. We informed parents upon enrollment that the outcomes survey was divvied into smaller aliquots except for annual follow-up to proactively manage expectations. We intentionally parsed the study instruments into different intervals to decrease response burden while creating a trajectory for each outcome metric (without a predesignated importance of obtaining certain instruments at specific time intervals). We asked parents to complete any previously missed follow-up instruments at the next follow-up interval, allowing for more complete data and recognizing that parents may differ in availability and willingness to participate. 

Our low response rates at 2 and 3 years are not surprising and represent the challenges inherent to long-term remote follow-up. The ability to maintain a meaningful relationship or a relationship without a face-to-face connection is difficult. As more time ensues from PICU discharge to remote follow-up, parents may be less inclined to participate in an activity in which the patient or family does not receive a tangible health benefit (i.e., medical or neurodevelopmental evaluation or therapies). Financial incentive for participation may need to account for this attrition. Future efforts to increase retention rates in PICU follow-up programs would likely benefit from clinician and family education as well as care coordination; education regarding the importance of long-term follow-up and scheduling of these visits prior to discharge have yielded higher rates of neonatal follow-up clinic participation [50]. 

Our study had important limitations. Because our study design required in-person parent consent and completion of baseline study measures, only 21% of all eligible children were enrolled. We were not able to evaluate the demographic and clinical characteristics of children who did not enroll to understand if they were systematically different from our overall patient population. Our enrollment rates reflect the socioeconomic challenges (single-parent households, competing sibling care demands, inflexible work requirements, poverty, lack of transportation, etc.) that preclude regular parent presence at the bedside. Additionally, we excluded children who were wards of the state or whose parents were primarily non-English speakers due to logistical challenges related to changes in custody after discharge and coordination of interpreter services, parents, and study teams for telephone follow-up. Therefore, our findings may have limited generalizability and may not delineate the additional challenges presented by linguistic barriers. However, our study population was representative of our overall patient population with regard to demographic and clinical characteristics except for older age. Any resultant bias due to an older patient population is unclear, as our EHR does not contain data to determine neurodevelopmental vulnerability. Most parents reported that they had not answered a prior contact attempt because they were busy and did not describe feeling overwhelmed by touchpoints. Directly measuring time burden for parents would provide insight into potential barriers to study completion and inform future work. Semistructured interviews with families to assess the feasibility or perceived burden of participating in longitudinal follow-up studies would further inform efforts to increase recruitment and retention for PICU survivorship studies. Studies indicate that a sense of partnership and family-focused interventions are important facilitators for neonatal follow-up [51,52].

In this manuscript, we report details regarding follow-up methodology and did not present the outcomes at long-term follow-up. In the nascent field of PICU survivorship, providing a methodologic “roadmap” to conduct long-term outcome research is essential. This methodology serves as a starting point for investigators to address the feasibility challenges associated with follow-up studies in a heterogenous population without a common medical home. We did not present methodology and results as a single manuscript because publication page limits would necessitate an abbreviated focus on methodology without the level of detail we were able to provide here.

## 5. Conclusions

Our results suggest that a multidimensional longitudinal follow-up for a heterogeneous group of children who survive pediatric critical illness is feasible. We identified logistical and resource challenges with enrollment. Frequent attempts to connect with families with flexibility in scheduling remote follow-up, financial incentivization, and minimization of response burden are key elements for retaining a cohort of PICU survivors. Ultimately, understanding the multidimensional scope and trajectories of long-term outcomes facing PICU survivors will allow us to develop and implement interventions that will maximize quality of life and overall functioning for children who experience critical illness.

## Figures and Tables

**Figure 1 children-09-01041-f001:**
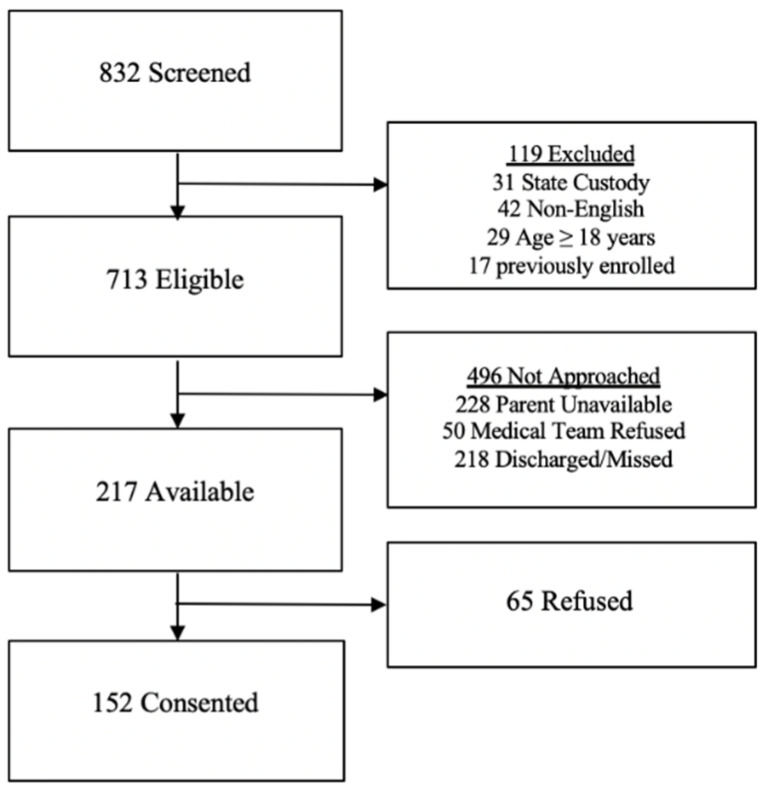
Study enrollment.

**Figure 2 children-09-01041-f002:**
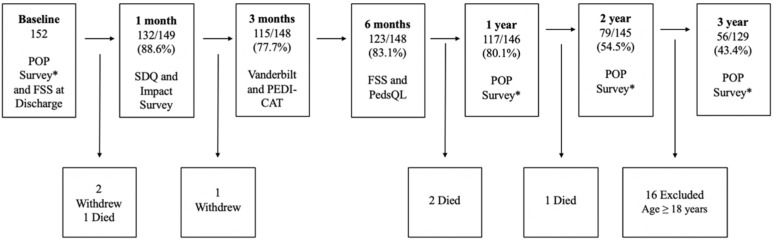
Cohort retention and follow-up among PICU survivors. * POP survey = impact survey, functional status scale (FSS), Pediatric Evaluation of Disability Inventory Computer Adaptive Test (PEDI-CAT), Pediatric Quality of Life Inventory^TM^ (PedsQL), Strengths and Difficulties Questionnaire (SDQ), and Vanderbilt Assessment Scales for Attention Deficit-Hyperactivity Disorder (Vanderbilt); PEDI-CAT was not included at 1- and 2-year follow-up.

**Table 1 children-09-01041-t001:** Demographic and clinical characteristics of study cohort.

Characteristic	Enrollment/Discharge (n = 152)
Age in years, mean (SD)	5.2 (5.2)
Sex, n (%)	
Male	89 (58.6)
Female	63 (41.4)
Race, n (%)	
Black	87 (57.2)
White	51 (33.6)
Other	14 (9.2)
Ethnicity, n (%)	
Hispanic	14 (9.2)
Non-Hispanic	138 (90.8)
Primary Diagnosis, n (%)	
Pulmonary	60 (39.5)
Surgical	33 (21.7)
Neurologic	13 (8.6)
Trauma	12 (7.9)
Infection	11 (7.2)
Other	23 (15.1)
Admission PSOFA, mean (SD) °	4.6 (2.4)
Admission PRISM, mean (SD) °	9.2 (4.5)
Median length of PICU stay °, days (IQR)	3.0 (1.7–8.1)
Median length of Hospital stay °, days (IQR)	4 (3–9)

° admission pSOFA and PRISM calculated as maximum scores within 24 h of PICU admission, hospital LOS calculated as full days during hospitalization encounter; PICU stay determined by time of first and last vital-sign recordings.

**Table 2 children-09-01041-t002:** Characteristics of successful follow-up attempts.

	Follow-Up Interval					
	1 Month (n = 132)	3 Months (n = 115)	6 Months (n = 123)	1 Year (n = 117)	2 Years (n = 79)	3 Years (n = 56)
Number of follow-up attempts, median (IQR)	2 (1–4)	2 (1–3)	2 (1–3)	3 (2–6)	2 (1–4)	4 (3–5)
Day						
Weekday completion, n (%)	127 (96.2)	112 (97.4)	115 (93.5)	104 (88.9)	69 (87.3)	48 (85.7)
Weekend completion, n (%)	5 (3.8)	3 (2.6)	8 (6.5)	13 (11.1)	10 (12.7)	8 (14.3)
Time *						
Daytime completion, n (%)	116 (89.2)	99 (91.7)	94 (82.4)	88 (76.5)	57 (72.2)	38 (67.9)
Evening completion, n (%)	14 (10.8)	9 (8.3)	20 (17.35)	27 (23.5)	22 (27.8)	18 (32.1)
Days to completion after follow-up eligible, median (IQR)	9 (0–34.3)	26 (10.5–68)	23 (6.5–49)	28 (9–65)	12 (2–28)	38.5 (17.3–124.8)
Completed at later follow-up interval, n (%)	19 (14.5)	21 (19.3)	1 (0.8)	N/A	N/A	N/A

* Time follow-up was not captured for a minority of patients (<10% at all intervals).

## Data Availability

The data presented in this study are available on request from the corresponding author. The data are not publicly available.

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
