# Peer review of "PICU Survivorship: Factors Affecting Feasibility and Cohort Retention in a Long-Term Outcomes Study"

_children, 2022, doi:10.3390/children9071041_

Round 1

Reviewer 1 Report

The current manuscript by Sobotka et al. addresses an important question of clinical relevance regarding the engagement and retention of high-risk infants for “long-term” follow-up post-PICU discharge. Strengths of this study include a prospectively recruited contemporary cohort, with diverse diagnoses (39.5% pulmonary, 21.7% surgical, 8.6% neurologic, 7.9% trauma, 7.2% infection, and 15.1% other), and a multidimensional neurodevelopmental follow-up protocol. Despite these strengths, this study was significantly limited by its research design and methodological characteristics, as outlined below, as well as the presentation of findings in the manuscript.

While the manuscript is generally well-written, the Introduction fails to fully justify the rationale of the current study. Although it provides a context for the critical need for neurodevelopmental follow-up in this high-risk population, there is a limited acknowledgment of rates of follow-up as documented in existing studies. Further, it would have been helpful to draw on findings from other high-risk infant populations such as preterm birth or congenital heart disease along with highlighting well-known risk factors for loss to follow-up.

The last paragraph of the Introduction is particularly unclear as it primarily describes the larger cohort study, except for the final sentence which has relevance to this study. I strongly suggest rewriting this paragraph, clearly identifying the study aims and core metrics to assess those outcomes. Throughout the manuscript, I struggled to understand the authors’ metrics for assessing the two core outcomes, i.e., “feasibility of longitudinal multidimensional outcomes assessment, and methods to promote cohort retention.” Unfortunately, this lack of clarity is a major weakness of the entire manuscript. For example, in the aims (line 62), the authors refer to a “novel outcomes portfolio”, which is vague, confusing, and unnecessary.

As I understand, the authors have used “retention rate” as the marker of feasibility, which represents only one of the many aspects of feasibility. It would have been helpful to assess feasibility and burden from the parents’ and providers’ perspectives using both qualitative and quantitive methodologies.

It would be helpful for the readers to know a bit more about the PICU at this center (Comer Children’s Hospital at the University of Chicago) to determine the representativeness of the sample to the larger regional, national, and international PICU population.

I also failed to understand the research design within the context of the study aims. Specifically, the reason is not convincing enough for excluding 41 non-English speaking families and 31 children in state custody. I respectfully disagree with their justification unless this is an exploratory/pilot study. Unfortunately, this is a major limitation (and almost a fatal flaw), as these are the two highest-risk subgroups vulnerable to loss to follow-up, as documented in the literature.

Similarly, parental presence at the bedside for enrollment as inclusion is not justified. For a well-designed study, given the study aims, it would have been beneficial if the authors had considered developing innovative methods for recruiting this subgroup of parents. Again, in my opinion, this is a missed opportunity.

In line 80, the authors have identified that “Bedside nurses facilitated opportunities for consent by notifying the study team when parents became available.” Please clarify if the nurses were “trained” to identify the best opportunities to facilitate successful recruitment.

Please provide a relevant description of the development of the Healthcare and Neurodevelopmental Profile and Family Impact Survey. Specifically, please describe the steps involved in the selection of items, the total number of items, revision after cognitive testing, etc. Also, please provide the psychometrics of the final survey items and the full survey in the Appendix.

Also, the rationale for including both SDQ and Vanderbilt scale remains unclear.

Please provide relevant psychometric data for all included instruments.

I was surprised that the manuscript had very limited consideration for family social characteristics despite the well-established finding as a major risk factor for loss to follow-up. I strongly suggest including a range of family social risk variables in the sample characteristics description.

As the overall enrollment rate was very low, it is critical to characterize the non-enrolled participants to understand their risk profiles, based on available data. It would be helpful to understand the clinical and social risk profiles of the 65 refusals, 218 discharged/missed, and 50 medical team refusals.

In the Methods, please clarify if participants were concurrently enrolled in any other clinical or research follow-up programs, as that would impact their participation.

It is also important to describe the extent of awareness and knowledge of the participants (and the providers) regarding the critical need and benefit of attending long-term follow-up post-PICU discharge. Health literacy is an important variable that determines attendance, and so does the nature of discharge counseling and referral practices.

While the sample size may not permit, if feasible, the authors may consider a subgroup analysis of different diagnoses. I believe the findings would be interesting for the readers.

The Discussion needs major revision. Specifically, the authors need to contextualize current findings within available literature,  avoid overstating the significance and “novelty” of the current findings, and most important acknowledge thoroughly the limitations of the study design and methods.

In line 257, the authors state, “Thus, our consent rate may reflect a willingness to participate in interventions to improve long-term outcomes.” I am not sure how or why?

The authors need to carefully review their references throughout the manuscript. Specifically, several statements omitted references when it was needed, and in other instances, more references were included than needed.

Finally, I strongly recommend the authors revise the manuscript title to align more closely with the limited scope of the study outcomes (as described above).

Author Response

Response to Reviewer 1 Comments

The current manuscript by Sobotka et al. addresses an important question of clinical relevance regarding the engagement and retention of high-risk infants for “long-term” follow-up post-PICU discharge. Strengths of this study include a prospectively recruited contemporary cohort, with diverse diagnoses (39.5% pulmonary, 21.7% surgical, 8.6% neurologic, 7.9% trauma, 7.2% infection, and 15.1% other), and a multidimensional neurodevelopmental follow-up protocol. Despite these strengths, this study was significantly limited by its research design and methodological characteristics, as outlined below, as well as the presentation of findings in the manuscript.

Point 1: While the manuscript is generally well-written, the Introduction fails to fully justify the rationale of the current study. Although it provides a context for the critical need for neurodevelopmental follow-up in this high-risk population, there is a limited acknowledgment of rates of follow-up as documented in existing studies. Further, it would have been helpful to draw on findings from other high-risk infant populations such as preterm birth or congenital heart disease along with highlighting well-known risk factors for loss to follow-up.

Response 1: In response to Point 1 and Point 3 below, we have presented data for current rates of enrollment and follow-up from existing studies to provide a rationale for the current study in the introduction (see bold):

“However, prior to adaption, the factors governing feasibility of implementing core outcome sets must be considered. Among the 407 studies included in a recent scoping review of PICU long-term outcomes studies, enrollment in observational PICU survivorship studies varied from 61-100%, with even lower enrollment in qualitative studies ranging from 20-90%. (REFERENCE 28, Maddux et al). Retention at long-term follow-up was also highly variable, ranging from 68-100%. (REFERENCE 28, Maddux et al). In this prospective study piloting the use of multidimensional outcome measures among a heterogeneous group of children who experience critical illness, we set out to understand the factors that affect parental participation in a longitudinal follow-up study. Here, we report our experience with regard to feasibility and cohort retention in a PICU follow-up study through 3 years after hospital discharge.”

In the 2nd paragraph of the Introduction, we have also acknowledged a key difference between other high-risk infant populations and the PICU population that influence loss to follow-up (see bold).

“Our understanding of the extent of PICS-p is obscured by the variability in duration of follow-up, patient populations studied, and outcome measures. Importantly, PICU survivors do not have a medical home, distinguishing them from preterm infants or children with congenital heart disease who have dedicated follow-up clinics, potentially placing them at increased risk for loss to follow-up.

In the Discussion, we have also acknowledged lessons learned from neonatal follow-up:

“Our low response rates at 2 and 3 years are not suprising and represent the challenges inherent to long-term remote follow-up. The ability to maintain a meaningful relationship or a relationship without a face-to-face connection is difficult. As more time ensues from PICU discharge to remote follow-up, parents may be less inclined to participate in an activity in which the patient or family does not receive a tangible health benefit (i.e. medical or neurodevelopmental evaluation or therapies). Financial incentive for participation may need to account for this attrition. Future efforts to increase retention rates in PICU follow-up programs would likely benefit from clinician and family education as well as care coordination; education regarding the importance of long-term follow-up and scheduling of these visits prior to discharge have yielded higher rates of neonatal follow-up clinic participation. (Reference: Brachio SS, Farkouh-Karoleski C, Abreu A, Zygmunt A, Purugganan O, Garey D. Improving Neonatal Follow-up: A Quality Improvement Study )Analyzing In-hospital Interventions and Long-term Show Rates. Pediatr Qual Saf. 2020 Oct 23;5(6):e363. doi: 10.1097/pq9.0000000000000363. PMID: 33575523; PMCID: PMC7870232 .)

Our study had important limitations. Because our study design required in-person parent consent and completion of baseline study measures, only 21% of all eligible children were enrolled. We were not able to evaluate the demographic and clinical characteristics of children who did not enroll to understand if they were systematically different from our overall patient population. Our enrollment rates reflect the socioeconomic challenges (single parent households, competing sibling care demands, inflexible work requirements, poverty, lack of transportation, etc.) that preclude regular parent presence at the bedside. Additionally, we excluded children who were wards of the state or whose parents were primarily non-English speakers due to logistical challenges related to changes in custody after discharge and coordination of interpreter services, parents, and study team for telephone follow-up. Therefore, our findings may have limited generalizability and may not delieneate the additional challenges presented by linguistic barriers. However, our study population was representative of our overall patient population with regard to demographic and clinical characteristics except for older age. Any resultant bias due to an older patient population is unclear as our EHR does not contain data to determine neurodevelopmental vulnerability. Most parents reported that they had not answered a prior contact attempt because they were busy and did not describe feeling overwhelmed by touchpoints. Directly measuring time burden for parents would provide insight into potential barriers to study completion and inform future work. Studies indicate that a sense of partnership and family-focused interventions are important faciltators for neonatal follow-up. (Reference: Ballantyne M, Benzies K, Rosenbaum P, Lodha A. Mothers' and health care providers' perspectives of the barriers and facilitators to attendance at Canadian neonatal follow-up programs. Child Care Health Dev. 2015 Sep;41(5):722-33. doi: 10.1111/cch.12202. Epub 2014 Oct 1. PMID: 25272331.) (Reference: Ballantyne M, Stevens B, Guttmann A, Willan AR, Rosenbaum P. Maternal and infant predictors of attendance at Neonatal Follow-Up programmes. Child Care Health Dev. 2014 Mar;40(2):250-8. doi: 10.1111/cch.12015. Epub 2013 Jan 7. PMID: 23294101.) Semi-structured interviews with families to assess the feasibility or perceived burden of participating in longitudinal follow-up studies would further inform efforts to increase recruitment and retention for PICU survivorship studies.”

Point 2: The last paragraph of the Introduction is particularly unclear as it primarily describes the larger cohort study, except for the final sentence which has relevance to this study. I strongly suggest rewriting this paragraph, clearly identifying the study aims and core metrics to assess those outcomes. Throughout the manuscript, I struggled to understand the authors’ metrics for assessing the two core outcomes, i.e., “feasibility of longitudinal multidimensional outcomes assessment, and methods to promote cohort retention.” Unfortunately, this lack of clarity is a major weakness of the entire manuscript. For example, in the aims (line 62), the authors refer to a “novel outcomes portfolio”, which is vague, confusing, and unnecessary.

Response 2: Please provide your response for Point 2. (in red)

We have revised the last paragraph of the Introduction in response to the reviewer’s concerns:

“However, prior to adaption, the factors governing feasibility of implementing core outcome sets must be considered. Among the 407 studies included in a recent scoping review of PICU long-term outcomes studies, enrollment in observational PICU survivorship studies varied from 61-100%, with even lower enrollment in qualitative studies ranging from 20-90%. (REFERENCE 28, Maddux et al). Retention at long-term follow-up was also highly variable, ranging from 68-100%. (REFERENCE 28, Maddux et al). In this prospective study piloting the use of multidimensional outcome measures among a heterogeneous group of children who experience critical illness, we set out to understand the factors that affect parental participation in a longitudinal follow-up study. Here, we report our experience with regard to feasibility and cohort retention in a PICU follow-up study through 3 years after hospital discharge.”

Point 3: As I understand, the authors have used “retention rate” as the marker of feasibility, which represents only one of the many aspects of feasibility. It would have been helpful to assess feasibility and burden from the parents’ and providers’ perspectives using both qualitative and quantitive methodologies.

Response 3: We agree that participation and retention in a study are among many aspects of feasibility. We highlighted other aspects of feasibility such as timing (daytime vs. evening and weekday vs. weekend) in Table 2 as well as method of preferred contact (telephone vs email) and mode of administration (telephone vs email) in the Results and Discussion.

Although we were unable to quantitatively and qualitatively assess the time burden experienced by families, we did note that nearly 20% of families were willing to complete previous surveys at a subsequent interval. In the Limitations paragraph, we acknowledged that measuring time burden would be informative. In response to the reviewer, we additionally acknowledge the lack of quantitative and qualitative assessment other factors governing feasibility as an important limitation:

“Our study had important limitations. Because our study design required in-person parent consent and completion of baseline study measures, only 21% of all eligible children were enrolled. We were not able to evaluate the demographic and clinical characteristics of children who did not enroll to understand if they were systematically different from our overall patient population. Our enrollment rates reflect the socioeconomic challenges (single parent households, competing sibling care demands, inflexible work requirements, poverty, lack of transportation, etc.) that preclude regular parent presence at the bedside. Additionally, we excluded children who were wards of the state or whose parents were primarily non-English speakers due to logistical challenges related to changes in custody after discharge and coordination of interpreter services, parents, and study team for telephone follow-up. Therefore, our findings may have limited generalizability. However, our study population was representative of our overall patient population with regard to demographic and clinical characteristics except for older age. Any resultant bias due to an older patient population is unclear as our EHR does not contain data to determine neurodevelopmental vulnerability. Most parents reported that they had not answered a prior contact attempt because they were busy and did not describe feeling overwhelmed by touchpoints. Directly measuring time burden for parents would provide insight into potential barriers to study completion and inform future work. Studies indicate that a sense of partnership and family-focused interventions are important faciltators for neonatal follow-up. (Reference: Ballantyne M, Benzies K, Rosenbaum P, Lodha A. Mothers' and health care providers' perspectives of the barriers and facilitators to attendance at Canadian neonatal follow-up programs. Child Care Health Dev. 2015 Sep;41(5):722-33. doi: 10.1111/cch.12202. Epub 2014 Oct 1. PMID: 25272331.) (Reference: Ballantyne M, Stevens B, Guttmann A, Willan AR, Rosenbaum P. Maternal and infant predictors of attendance at Neonatal Follow-Up programmes. Child Care Health Dev. 2014 Mar;40(2):250-8. doi: 10.1111/cch.12015. Epub 2013 Jan 7. PMID: 23294101.) Semi-structured interviews with families to assess the feasibility or perceived burden of participating in longitudinal follow-up studies would further inform efforts to increase recruitment and retention for PICU survivorship studies.”

Point 4: It would be helpful for the readers to know a bit more about the PICU at this center (Comer Children’s Hospital at the University of Chicago) to determine the representativeness of the sample to the larger regional, national, and international PICU population.

Response 4:

We added the following to the Results, Cohort Characteristics:

“The University of Chicago Comer Children’s Hospital is an urban, academic, tertiary care center that admits medical and surgical pediatric patients. Demographic and clinical characteristics of the study cohort are presented in Table 1. The median (range) age of the children was 5.2 (0-17) years, and 58.6% were male. Participants were 57.2% African-American (n=87), 33.6% Caucasian (n=51), 9.2% “Other” (n=14), and 9.2% Hispanic (n=14). At the time of study enrollment, our typical patient population was 54.2% male, 59.8% African-American, 26.7% White, and 24% surgical patients. Our overall PICU median length of stay was 2.9 days. Our study patients were similar to all patients admitted to the PICU during the study enrollment period with regard to sex, race, ethnicity, severity of illness, and length of stay. Study participants were older than the overall PICU population. (Supplemental Table 1)”

Point 5: I also failed to understand the research design within the context of the study aims. Specifically, the reason is not convincing enough for excluding 41 non-English speaking families and 31 children in state custody. I respectfully disagree with their justification unless this is an exploratory/pilot study. Unfortunately, this is a major limitation (and almost a fatal flaw), as these are the two highest-risk subgroups vulnerable to loss to follow-up, as documented in the literature.

Response 5:

We understand the reviewer’s concerns but could not include children in state custody for parent informant surveys that require day to day interactions for the responses. We also could not include non-English speaking families due to the lack of infrastructure at our institution to support interpreter use for research purposes. In response to the reviewer’s concerns, we additionally acknowledged the lack of generalizability in the Limitations:

“Our study had important limitations. Because our study design required in-person parent consent and completion of baseline study measures, only 21% of all eligible children were enrolled. We were not able to evaluate the demographic and clinical characteristics of children who did not enroll to understand if they were systematically different from our overall patient population Our enrollment rates reflect the socioeconomic challenges (single parent households, competing sibling care demands, inflexible work requirements, poverty, lack of transportation, etc.) that preclude regular parent presence at the bedside. Additionally, we excluded children who were wards of the state or whose parents were primarily non-English speakers due to logistical challenges related to changes in custody after discharge and coordination of interpreter services, parents, and study team for telephone follow-up. Therefore, our findings may have limited generalizability and may not delineate the additional challenges presented by linguistic barriers. However, our study population was representative of our overall patient population with regard to demographic and clinical characteristics except for older age. Any resultant bias due to an older patient population is unclear as our EHR does not contain data to determine neurodevelopmental vulnerability. Most parents reported that they had not answered a prior contact attempt because they were busy and did not describe feeling overwhelmed by touchpoints. Directly measuring time burden for parents would provide insight into potential barriers to study completion and inform future work. Studies indicate that a sense of partnership and family-focused interventions are important faciltators for neonatal follow-up. (Reference: Ballantyne M, Benzies K, Rosenbaum P, Lodha A. Mothers' and health care providers' perspectives of the barriers and facilitators to attendance at Canadian neonatal follow-up programs. Child Care Health Dev. 2015 Sep;41(5):722-33. doi: 10.1111/cch.12202. Epub 2014 Oct 1. PMID: 25272331.) (Reference: Ballantyne M, Stevens B, Guttmann A, Willan AR, Rosenbaum P. Maternal and infant predictors of attendance at Neonatal Follow-Up programmes. Child Care Health Dev. 2014 Mar;40(2):250-8. doi: 10.1111/cch.12015. Epub 2013 Jan 7. PMID: 23294101.) Semi-structured interviews with families to assess the feasibility or perceived burden of participating in longitudinal follow-up studies would further inform efforts to increase recruitment and retention for PICU survivorship studies.”

If we are able to demonstrate vulnerability among a population with less risk of loss to follow-up, we believe this will underscore how non-English speaking families and children in state custody are likely at higher risk of vulnerability.

Point 6: Similarly, parental presence at the bedside for enrollment as inclusion is not justified. For a well-designed study, given the study aims, it would have been beneficial if the authors had considered developing innovative methods for recruiting this subgroup of parents. Again, in my opinion, this is a missed opportunity.

Response 6: We agree with the reviewer. However, this study was initiated prior to changes at our institutional IRB which allowed for greater flexibility in obtaining parental consent. At the time of study enrollment, verbal or telephone consent for participation in research studies was not permissable in our institution. As such, parental presence at the bedside for enrollment was a mandatory inclusion criterion.  

Point 7: In line 80, the authors have identified that “Bedside nurses facilitated opportunities for consent by notifying the study team when parents became available.” Please clarify if the nurses were “trained” to identify the best opportunities to facilitate successful recruitment.

Response 7:

We did not specifically train nurses to identify best opportunites to faciliate successful recruitment as they were not part of the study team. However, nurses were able to alert the study team to when parents became available. Subsequently, trained study coordinators would discuss whether it was appropriate to approach the parents with the clinical team prior to attempting to recruit and enroll.

Point 8: Please provide a relevant description of the development of the Healthcare and Neurodevelopmental Profile and Family Impact Survey. Specifically, please describe the steps involved in the selection of items, the total number of items, revision after cognitive testing, etc. Also, please provide the psychometrics of the final survey items and the full survey in the Appendix

Response 8: Our research team developed the Impact Survey in order to capture aspects of health care utilization and the impact of the critical illness on overall family functioning not otherwise covered by existing tools. We had described the creation of this survey previously in our manuscript, and added now added detail:

 “The Impact Survey, designed by the study investigators, captures patients’ baseline health utilization and the family impact of the child’s illness. Items were cognitively tested with developmental and behavioral pediatric clinic patients prior to study initiation. The final survey length depended on an individual’s responses, but included about 30 questions. The Impact Survey includes modified questions from the National Survey of Children with Special Health Care Needs[39] regarding the child’s health (e.g., “What kind of place does your child go to when he/she is sick or you need advice about his/her health?”). In addition, questions capture the child’s access to and utilization of neurodevelopmental support services (e.g., “Does your child have an IEP (Individualized Educational Program)?”). Family impact is also assessed (e.g., “Has your child’s health conditions caused financial problems for your family?”).”

We have included in a correspondence with the editorial team a PDF of this full survey.

Point 9: Also, the rationale for including both SDQ and Vanderbilt scale remains unclear.

Response 9: We apologize if this was unclear. The SDQ is a measure of pro-social and psychopathology, with subcategories that include overall difficulties, emotional problems, conduct problems, hyperactivity, peer problems, and prosocial problems. It is a brief screening which does not provide specific diagnoses. The Vanderbilt is specifically focused on ADHD symptomatology and can be used to diagnose this disorder as well as measure severity. We used both tools as there is not yet a standard tool for the assessment of psychosocial functioning after PICU admission, and the behavioral and emotional manifestations of PICS-p may very widely. We will be happy to add this additional detail into the manuscript as directed by the journal editorial and reviewer team.

Point 10: Please provide relevant psychometric data for all included instruments.

Response 10:

We had provided references for the validation for the included instruments. We included avaliable data for the PEDI-CAT, PedsQL, SDQ, and Vanderbilt.

“The PEDI-CAT includes 15 items in each of four domains (daily activities, mobility, social/cognitive, and responsibility) to determine the degree of functioning in each category compared to same-aged peers. Adaptive testing maximizes information gathering while minimizing response burden[35].

The PedsQL is a multi-dimensional tool intended to quantify health-related quality of life in healthy and ill children up to 18 years of age. The PedsQL consists of 23 items in five domains (physical, psychosocial, emotional, social, and school)[36]. This measure was administered to all parents with versions tailored to the age range of the child: 1-12 months, 13-24 months, 2-4 years, 5-7 years, 8-12 years, or 13-18 years. The population mean for parent-proxy report 81.3 and standard deviation 15.9 for healthy children. (Reference: Varni JW, Burwinkle TM, Seid M, Skarr D. The PedsQL 4.0 as a pediatric population health measure: feasibility, reliability, and validity. Ambul Pediatr. 2003 Nov-Dec;3(6):329-41. doi: 10.1367/1539-4409(2003)003<0329:tpaapp>2.0.co;2. PMID: 14616041.)

The SDQ is a brief behavioral screening survey to assess pro-social behavior and psychopathology of children and adolescents aged 4-17 years[37]. Normative parent SDQ scores for U.S. children are 0-11 (for low difficulties, 12-15 for medicum difficulties, and 16-40 for hight difficulties. (Reference: Bourdon KH, Goodman R, Rae DS, Simpson G, Koretz DS. The Strengths and Difficulties Questionnaire: U.S. normative data and psychometric properties. J Am Acad Child Adolesc Psychiatry. 2005 Jun;44(6):557-64. doi: 10.1097/01.chi.0000159157.57075.c8. PMID: 15908838.)

The Vanderbilt is a tool that parallels signs and symptoms from DSM-IV to help healthcare professionals diagnose ADHD in children[38]. The Vanderbilit was administered to parents of children at least 5 years of age. Normative data for total Attention Deficit Hyperactivity Disorder (ADHD) score, ADHD inattentive, ADHD hyperactive, Oppositional Defiant Disorder (ODD), Conduct Disorder, and Anxiety/Depression are 3.4, 1.6, 1.8, 1.1, 0.5, and 0.4, respectively. (Reference: Nathan P Anderson, BA, Jamie A Feldman, BA, David J Kolko, PhD, Paul A Pilkonis, PhD, Oliver Lindhiem, PhD, National Norms for the Vanderbilt ADHD Diagnostic Parent Rating Scale in Children, Journal of Pediatric Psychology, 2022;, jsab132, https://doi.org/10.1093/jpepsy/jsab132)”

Point 11: I was surprised that the manuscript had very limited consideration for family social characteristics despite the well-established finding as a major risk factor for loss to follow-up. I strongly suggest including a range of family social risk variables in the sample characteristics description.

Response 11:

Thank you for raising this important point. We have added the following to the third paragraph in the Discussion (see bold).

“While only 21% of all eligible families enrolled, the majority of families that were available to the study team agreed to participate. This high consent rate of 70% indicates a subgroup of patients that is willing to participate in studies of long-term follow-up. Our ability to enroll a smaller number of families compared to those who consented, reflecting loss to follow-up, may be explained, in part, by the demographic and social characteristics associated with risk of poor outcomes. For example, young caregiver age, caregiver language barriers, presence of social supports, transportation challenges, or caregiver intellectual disability are risk factors for adverse PICS-p outcomes. (Reference: Woodruff AG, Choong K. Long-Term Outcomes and the Post-Intensive Care Syndrome in Critically Ill Children: A North American Perspective. Children (Basel). 2021 Mar 24;8(4):254. doi: 10.3390/children8040254. PMID: 33805106; PMCID: PMC8064072.) These same factors may inhibit access to care, precluding participation in routine clinical or research-related follow-up.”

Point 12: As the overall enrollment rate was very low, it is critical to characterize the non-enrolled participants to understand their risk profiles, based on available data. It would be helpful to understand the clinical and social risk profiles of the 65 refusals, 218 discharged/missed, and 50 medical team refusals.

Response 12:

We agree but unfortunately were not able to collect data on participants who did not enroll to understand if they were systematically different. We have acknowledged this in our Limitations paragraph (see bold).

“Our study had important limitations. Because our study design required in-person parent consent and completion of baseline study measures, only 21% of all eligible children were enrolled. We were not able to evaluate the demographic and clinical characteristics of child who did not enroll to understand if they were systematically different from our overall patient population. Our enrollment rate reflects…

Point 13: In the Methods, please clarify if participants were concurrently enrolled in any other clinical or research follow-up programs, as that would impact their participation.

Response 13: Participants were not concurrently enrolled in any other clinical or research follow-up programs as this was the only follow-up program for PICU patients in our institution. We have clarified in the Methods, Patients section:

“Patients were enrolled from the PICU at Comer Children’s Hospital at the University of Chicago over 15 months (June 2017-August 2018) to account for the seasonal variation inherent in pediatric critical illness and injury. All children aged 0 to 17 years admitted to the PICU were eligible; parental presence at the bedside was necessary for consent and enrollment. No competing clinical or research follow-up programs were in existence at the time of study enrollment. Children were excluded if they were under state custody due to the logistical challenges of obtaining consent from the state for participation in a research study as well as potential flux in custodial situations at follow-up intervals. Non-English speakers were excluded because some instruments were only available in English with limited ability for translation during remote (telephone or e-mail) follow-up. Informed consent was obtained in person from parent(s)/guardians (hereafter, referred to as “parent(s)”), and assent was obtained from children aged 12 years or older. The protocol was approved by the Institutional Review Board at the University of Chicago.

Point 14: It is also important to describe the extent of awareness and knowledge of the participants (and the providers) regarding the critical need and benefit of attending long-term follow-up post-PICU discharge. Health literacy is an important variable that determines attendance, and so does the nature of discharge counseling and referral practices.

Response 14:

The reviewer raises another important factor that may have influenced retention. We have also added this to the second to last paragraph in the Discussion (see bold):

Our low response rates at 2 and 3 years are not suprising and represent the challenges inherent to long-term remote follow-up. The ability to maintain a meaningful relationship or a relationship without a face-to-face connection is difficult. As more time ensues from PICU discharge to remote follow-up, parents may be less inclined to participate in an activity in which the patient or family does not receive a tangible health benefit (i.e. medical or neurodevelopmental evaluation or therapies). Financial incentive for participation may need to account for this attrition. Future efforts to increase retention rates in PICU follow-up programs would likely benefit from clinician and family education as well as care coordination; education regarding the importance of long-term follow-up and scheduling of these visits prior to discharge have yielded higher rates of neonatal follow-up clinic participation. (Reference: Brachio SS, Farkouh-Karoleski C, Abreu A, Zygmunt A, Purugganan O, Garey D. Improving Neonatal Follow-up: A Quality Improvement Study )Analyzing In-hospital Interventions and Long-term Show Rates. Pediatr Qual Saf. 2020 Oct 23;5(6):e363. doi: 10.1097/pq9.0000000000000363. PMID: 33575523; PMCID: PMC7870232 .)

Point 15: While the sample size may not permit, if feasible, the authors may consider a subgroup analysis of different diagnoses. I believe the findings would be interesting for the readers.

Response 15: We agree but sample size prohibited subgroup analyses as the reviewer acknowledged.

Point 16: The Discussion needs major revision. Specifically, the authors need to contextualize current findings within available literature,  avoid overstating the significance and “novelty” of the current findings, and most important acknowledge thoroughly the limitations of the study design and methods.

Response 16:

We have rewritten the Discussion in response the reviewer’s comments (specifically Points 3, 5, 11, 12, 14, and 17) and contextualized our findings in the context of available literature.

We have de-emphasized the “novelty” of our findings:

For example, the final sentence of the first paragraph of the Discussion was revised to:

“Our study demonstrates feasibility of implementing a core set of multidimensional outcomes that longitudinally assess the physical, cognitive, educational, social, emotional, behavioral, health-related quality of life, and family impact of pediatric critical illness in a heterogeneous patient population.”

Instead of:

“To the best of our knowledge, our study is the first of its kind to demonstrate feasibility implementing a core set of multidimensional outcomes that longitudinally assess the physical, cognitive, educational, social, emotional, behavioral, health-related quality of life, and family impact of pediatric critical illness in a heterogeneous patient population.”

Similarly, in the final sentence of the last paragraph of the Introduction was revised to:

“Here, we report our experience with regard to feasibility and cohort retention using a multidimensional outcomes portfolio among PICU survivors through 3 years after hospital discharge.”

Instead of:

Here, we report our experience with regard to feasibility and cohort retention using this novel outcomes portfolio among PICU survivors through 3 years after hospital discharge.

There is no other mention of novelty throughout the mansucript.

Point 17: In line 257, the authors state, “Thus, our consent rate may reflect a willingness to participate in interventions to improve long-term outcomes.” I am not sure how or why?

Response 17:

We have deleted this sentence.

Point 18: The authors need to carefully review their references throughout the manuscript. Specifically, several statements omitted references when it was needed, and in other instances, more references were included than needed.

Response 18:

We have reviewed the references and included more where needed.

Point 19: Finally, I strongly recommend the authors revise the manuscript title to align more closely with the limited scope of the study outcomes (as described above).

Response 19:

We have revised the title to:

“PICU Survivorship: Factors Affecting Feasibility and Cohort Retention in a Long-Term Outcomes Study”

Reviewer 2 Report

This is a well-written, well-researched manuscript, that I thoroughly enjoyed reading. Kudos to the authors! My (very minor) points are as follows:

  1. If the authors anticipated a three years period of follow-up, why did they enroll children up to the age of 17? It would have been obvious that some of them turned 18 in the meanwhile. Also, what would make them not elligible once they turned 18?
  2. I think the authors should further explore the difference of availability, if there is one, between parents who only had their child hospitalized once, and parents of children with multiple hospitalizations in the PICU. I would assume that the latter would be more readily available to answer the different questionnaires. Were there cases of multiple hospitalizations in this cohort? Maybe the authors don’t have the data, but in my oppinion, the issue requires a comment in the Discussion section.
  3. Table 2 is not entirely relevant in my opinion, but if the authors decide to keep it, it should be better fitted on the page. Also, I struggled with reading Figure 2.

Author Response

This is a well-written, well-researched manuscript, that I thoroughly enjoyed reading. Kudos to the authors! My (very minor) points are as follows:

Point 1: If the authors anticipated a three years period of follow-up, why did they enroll children up to the age of 17? It would have been obvious that some of them turned 18 in the meanwhile. Also, what would make them not elligible once they turned 18?

Response 1: We would like to thank the reviewer for pointing out that we had inaccurately described our study. Patients were eligible to continue in the follow-up portion of this study beyond the age of 18 years, however doing so did require a transition from parent consent to the child themselves providing consent. We found that the child serving as the informant was an obstacle to ongoing participation, and noted a drop out rate for children who became 18 years of age during this study observation period. We have revised the text below as follows:

Original text:

Three patients died during the study follow-up, and 4 patients withdrew consent. Sixteen children were excluded from 3 year follow-up because they were ³18 years of age and were no longer eligible.

Revised text:

Three patients died during the study follow-up, and 4 patients withdrew consent. Once children had turned age 18 they were the respondent and had to consent to participate. Sixteen children who became 18 years of age at the 3 year follow-up did not complete the final follow-up.

Point 2: I think the authors should further explore the difference of availability, if there is one, between parents who only had their child hospitalized once, and parents of children with multiple hospitalizations in the PICU. I would assume that the latter would be more readily available to answer the different questionnaires. Were there cases of multiple hospitalizations in this cohort? Maybe the authors don’t have the data, but in my oppinion, the issue requires a comment in the Discussion section.

Response 2: Thank you for this interesting point, and we agree that both duration of hospitalization and frequency of readmission would increase the likelihood that a child would be enrolled in our study.

We have added the following text to our limition section in the discussion:

Our study had important limitations. Because our study design required in-person parent consent and completion of baseline study measures, only 21% of all eligible children were enrolled. Our enrollment rates reflect the socioeconomic challenges (single parent household, competing sibling care demands, inflexible work requirements, poverty, lack of transportation, etc.) that preclude regular parent presence at the bedside. Parents of children who had longer hospitalizations, or who were readmitted during the study period, had a greater likelihood of having an opportunity to be recruited than other parents. Although this may reflect a selection bias, this cohort of families of children with complex critical illness are more likely to benefit from future PICU-follow-up supports to be informed by this study. Additionally, we excluded children who were wards of the state or whose parents were primarily non-English speakers due to logistical challenges related to changes in custody after discharge and coordination of interpreter services, parents, and study team for telephone follow-up. Therefore, our findings may have limited generalizability. 

Point 3: Table 2 is not entirely relevant in my opinion, but if the authors decide to keep it, it should be better fitted on the page. Also, I struggled with reading Figure 2.

Response 3:

We apologize, but due to the journal formatting instructions, when inserted into the word document appears longer than a page. We can also provide the table as a separate document for the editorial team. We defer to the editorial team to determine whether this Table is best suited as an Appendix.

Reviewer 3 Report

Sobotka et al. present an observational cohort study describing longitudinal followup for study purposes in a cohort of PICU patients who went home after admission. The manuscript is overall well written for content and the sections of the paper appear largely appropriately composed. Ethical review was performed.

Perhaps the biggest question is why are the results of the assessments not reported? Why exclude this information if it was the primary function of the interview process?

Author Response

Sobotka et al. present an observational cohort study describing longitudinal followup for study purposes in a cohort of PICU patients who went home after admission. The manuscript is overall well written for content and the sections of the paper appear largely appropriately composed. Ethical review was performed.

Point 1: Perhaps the biggest question is why are the results of the assessments not reported? Why exclude this information if it was the primary function of the interview process?

Response 1: Thank you for this inquiry. Due to this special edition, we were eager to respond to the invitation with a manuscript representing this body of work. However, at the time of the invitation, this comprehensive post-PICU outcomes study had outstanding 3-year follow-up which have now been completed and analysis is underway. We are in the process of completing a distinct manuscript which will focus on the outcomes over the full study period. Due to journal length restrictions, we thought a manuscript devoted to the feasibility and recruitment considerations would be of interest to diverse readership who engage with similar challenges in hospital cohort follow-up.

Round 2

Reviewer 1 Report

The authors have satisfactorily addressed all the comments of the reviewers.

Author Response

Thank you to the reviewer for comments and suggested revisions which have improved this manuscript.

Reviewer 3 Report

I appreciate the authors' response to the reviewers' comments. I think results of some sort after results are are in order, since data have been collected that information seems unusually absent. How would the authors address this in a later publication with the analysis since the impression of feasibility of the same study would be implicitly included in that paper? would this not seem like a double publication?

Author Response

Thank you to the reviewer for their consideration of this revised manuscript.

The field of PICU survivorship is a nascent area of research. The problems facing PICU outcomes research is two-fold: (1) retention without lack of standard clinical follow-up or outpatient medical home for PICU survivors; (2) standard strategies for studying longitudinal outcomes. For this reason, contributions on best strategies for retaining longitudinal cohorts is essential to the field. As such, we focused this manuscript on describing the methodology for recruiting and retaining this diverse cohort over several years of follow-up. Knowing that refining this methodology may be of benefit to other researchers of broad topics for PICU survivors, we aimed to describe our methods in greater detail than a manuscript also presenting results would enable. Because our population represented a broad PICU cohort, our retention strategies may be applicable and tailored to specific subpopulations.

The focus of the results paper will be describing trajectories of outcomes across a portfolio of measures of neurodevelopmental and overall health, with the primary objective to describe PICU follow-up outcomes over 3 years. Additionally, this manuscript will aim to define best outcome measures for future studies. In this future manuscript, reference to the study methodology will be limited to  broad strokes without detail on retention strategies.

We appreciate the reviewer and editorial teams for their ongoing consideration of this manuscript.

Round 3

Reviewer 3 Report

I appreciate the authors' responses to the reviewers' comments. As stated this is a plan for publication of two manuscripts with significant overlap and a likely redundant publication.

In the author instructions for this journal it states, "Authors should not unnecessarily divide their work into several related manuscripts, although short Communications of preliminary, but significant, results will be considered."

However no results of the actual study question are reported. Even if less is stated about the methods in the followup paper the experiment is identical between the two manuscripts. A paper with results, which the authors state is being drafted, will fully serve the function of this paper (to demonstrate follow-up is feasible) and therefore I have ethical concerns about publishing this paper as it is unnecessary. I defer to the editors on their judgement in this scenario.

Author Response

Our intent was to focus on the details of data acquisition and follow-up strategies in order to provide a methodological roadmap for future PICU long-term outcomes research.         We noted a gap in the literature on detailed methodology for recruiting and following-up this heterogenous critical care population, and therefore believe that these details provide a necessary contribution to the field. We will not be including these methodological details in a future publication, as we agree, that could be considered a duplicate publication if identical.              We respect and appreciate the concerns of this reviewer, and therefore defer to the editorial team to decide the outcome of this final revision submission.